# Three-Dimensional Bioprinting of an In Vitro Lung Model

**DOI:** 10.3390/ijms24065852

**Published:** 2023-03-19

**Authors:** Nádia Nascimento da Rosa, Julia Maurer Appel, Ana Carolina Irioda, Bassam Felipe Mogharbel, Nathalia Barth de Oliveira, Maiara Carolina Perussolo, Priscila Elias Ferreira Stricker, Lívia Rosa-Fernandes, Cláudio Romero Farias Marinho, Katherine Athayde Teixeira de Carvalho

**Affiliations:** 1Advanced Therapy and Cellular Biotechnology in Regenerative Medicine Department, Pelé Pequeno Príncipe Research Institute & Pequeno Príncipe Faculties, Curitiba 80240-020, Brazil; nadianr@gmail.com (N.N.d.R.); juliamappel@gmail.com (J.M.A.); anairioda@gmail.com (A.C.I.); bassamfm@gmail.com (B.F.M.); nathybarth03@gmail.com (N.B.d.O.); perussolo10@gmail.com (M.C.P.); priscilaeferreira@gmail.com (P.E.F.S.); 2Experimental Immunoparasitology Laboratory, Institute of Biomedical Sciences, University of São Paulo, São Paulo 05508-000, Brazil; liviarosa.f@gmail.com (L.R.-F.); marinho@usp.br (C.R.F.M.)

**Keywords:** mesenchymal stem cells, 3D bioprinting, lung, tissue engineering, human

## Abstract

In December 2019, COVID-19 emerged in China, and in January 2020, the World Health Organization declared a state of international emergency. Within this context, there is a significant search for new drugs to fight the disease and a need for in vitro models for preclinical drug tests. This study aims to develop a 3D lung model. For the execution, Wharton’s jelly mesenchymal stem cells (WJ-MSC) were isolated and characterized through flow cytometry and trilineage differentiation. For pulmonary differentiation, the cells were seeded in plates coated with natural functional biopolymer matrix as membrane until spheroid formation, and then the spheroids were cultured with differentiation inductors. The differentiated cells were characterized using immunocytochemistry and RT-PCR, confirming the presence of alveolar type I and II, ciliated, and goblet cells. Then, 3D bioprinting was performed with a sodium alginate and gelatin bioink in an extrusion-based 3D printer. The 3D structure was analyzed, confirming cell viability with a live/dead assay and the expression of lung markers with immunocytochemistry. The results showed that the differentiation of WJ-MSC into lung cells was successful, as well as the bioprinting of these cells in a 3D structure, a promising alternative for in vitro drug testing.

## 1. Introduction

The world is currently in a pandemic due to the outbreak of COVID-19. SARS-CoV-2, an enveloped RNA virus from bats, causes COVID-19. In December 2019, cases of infection by this virus in humans began in China, and in January 2020, the World Health Organization declared a state of international emergency [1,2,3].

Within this context, there is a significant search for new drugs to fight the disease, and there is also a need for in vitro models for preclinical tests of the drug’s cytotoxicity [4,5], especially because the main preclinical tests currently performed are on animals, a practice carried out since the time of Ancient Greeks such as Aristotle (384–322 BC) and Erasistratus (304–258 BC) [6]. In addition, in many cases, the results obtained by testing on animals are not reproducible in humans, causing harm to people and sacrificing the lives of many animals in vain [7,8].

One possibility to replace in vivo experiments are in vitro studies; cell culture is one of the most used in toxicology. However, the culture of only one cell type is still customarily carried out, kept in polystyrene bottles, with cells in suspension or in monolayer, which is a very different condition from what is found in the human body [9,10,11,12].

A new emerging technology is bioprinting associated with three-dimensional (3D) cell culture. 3D cell culture aims to reproduce the complexity of tissues in vitro. Particularly for lung tissue, commonly affected by the SARS-CoV-2 virus, it would be necessary, when performing toxicity tests, to analyze drugs from the perspective of a prototype that reproduces or replicates approaches to the alveolus. This pulmonary functional unit performs gas exchange, and may be a valuable alternative for in vitro cytotoxicity studies [11,12,13,14].

Berg et al. [11] have reported the development of a 3D bioprinted model for the study of influenza A infection. They used A549 cells, adenocarcinomic alveolar epithelial cells from humans, to establish a 3D bioprinted structure, which they infected with influenza A virus. Ng et al. [14] also used 549 cells, with MRC5 (human lung fibroblasts) and EA.hy926 (human endothelial cells), for the development of a triple-layered alveolar lung model that simulates the blood–air barrier system. 

However, a lung carcinoma cell line is not ideal for studying drug toxicity on normal cells. Therefore, the aim of this study was the development of a 3D lung model that can be used for in vitro cytotoxicity drug tests. To produce this model, Wharton’s jelly mesenchymal stem cells (WJ-MSC) were differentiated into lung cells.

## 2. Results

### 2.1. Mesenchymal Stem Cell Characterization

Flow cytometry results showed an average of 98.80% of the isolated cells with mesenchymal characteristics, that is, positive expression of the cell surface markers CD13 (aminopeptidase-N), CD73 (ecto-5-prime-nucleotidase), CD90 (Thy-1), and CD105 (endoglin), in addition to non-expression of the hematopoietic markers CD34 and CD45 (common leukocyte antigen). The immunogenicity was evaluated, and an average of 98.23% of non-hematopoietic cells presented histocompatibility, meaning they have expressive HLA-ABC and non-expressive HLA-DR. The histograms of sample C6 are shown in Figure 1.

All samples were successfully differentiated in the three lineages for the trilineage assay. The adipocytes presented lipid vacuoles stained with oil red O (Figure 2A), the osteoblasts showed calcium deposits stained with alizarin red (Figure 2C), and the presence of proteoglycans was possible to observe with alcian blue in the chondroblasts (Figure 2E).

### 2.2. Pulmonary Differentiation

#### 2.2.1. Immunocytochemistry

After differentiation, an immunocytochemistry assay was performed on the lung spheroids (Figure 3) and the cells that remained adhered to the plate during differentiation (Figure 4). In both cases, the cells expressed the proteins of secretory cells, ciliated cells, alveolar type 1 cells (AT1), and alveolar type 2 cells (AT2), showing successful differentiation. In addition, the pattern of each protein identified in the spheroids and isolated cells follows the literature [15,16,17,18,19]. The assay was also performed on the undifferentiated WJ-MSC as a negative control (Figure 5), and these cells did not express the evaluated proteins, confirming the differentiation. 

#### 2.2.2. Qualitative Reverse Transcription–Polymerase Chain Reaction (RT-PCR)

The RT-PCR of the beta-actin gene in the RNA samples treated with DNAse proves that the treatment successfully removed residual genomic DNA, since there were no amplification bands. In contrast, a non-treated RNA sample was used as a positive control. On the other hand, the amplification of the beta-actin gene in cDNA samples demonstrates that reverse transcription for cDNA production was effective (Figure 6).

The results of the RT-PCR analysis demonstrated that lung cells and undifferentiated WJ-MSC have very similar gene expression, and both cells expressed markers of secretory cells, AT1, and AT2. The gene expression of the cells used as a positive control (Calu-3) was also similar to that of WJ-MSC and differentiated lung cells (Figure 6). However, the quantification of the bands indicated that the differentiated pulmonary cells and positive control (Calu3) express these genes more than WJ-MSC (Appendix A), and these results were statistically significant at *p* < 0.001.

### 2.3. Bioink

To determine if the bioink was within the necessary standards, which are good printability and cytocompatibility, bioprinting tests were performed with the undifferentiated WJ-MSC. It was possible to observe that the bioink remained stable, as the structures were printed according to the model without deformations (Appendix A). It is also important to note that there was no precipitation of cells when the bioink was placed in the syringe, allowing the printing of a structure with a homogeneous cell concentration. Furthermore, a live/dead assay was performed after 24, 48, and 72 h of bioprinting, and the structures presented a cell viability of 92.15%, 92.02%, and 91.55%, respectively, proving the biocompatibility of the bioink (Figure 7 and Figure 8A).

### 2.4. 3D Bioprinting

#### 2.4.1. Live/Dead Assay

After bioprinting with the lung cells, the live/dead assay was performed again on the 3D structures to assess the viability of the differentiated cells and compare them with the undifferentiated WJ-MSC. The results obtained were similar to those of the WJ-MSC; the structures presented viability of 92.55%, 92.38%, and 91.66% after 24, 48, and 72 h, respectively, and there was no statistical difference between the cell types nor among the three times evaluated (Figure 8).

#### 2.4.2. Immunocytochemistry

Immunocytochemistry was performed on 3D lung structures for the same markers used for lung characterization. It was possible to observe that the cells continued to express the proteins of secretory cells, ciliated cells, AT1, and AT2, indicating that the 3D structure resembles lung tissue (Figure 9).

## 3. Discussion

Respiratory diseases are among the leading causes of morbidity and mortality worldwide, a situation that the COVID-19 pandemic has only exacerbated. For this reason, there is a significant interest in developing more efficient in vitro models that can mimic respiratory tissues and be used for cytotoxicity drug tests [3,20,21]. Therefore, this study’s objective was to develop a 3D lung model from differentiated WJ-MSC that can be used for in vitro studies of drug cytotoxicity.

WJ-MSCs were isolated from the human umbilical cord by the explant method. Their characterization followed the International Society for Cell Therapy, which defines that for a cell to be considered an MSC, it must show adherence to the plastic substrate, express the cell surface markers CD73 (ecto-5-prime-nucleotidase), CD90 (Thy-1), and CD105 (endoglin) through flow cytometry, not express CD34 and CD45 (common leukocyte antigen) surface markers through flow cytometry, have multipotent differentiation capacity, that is, ability to differentiate into adipocytes, chondrocytes, and osteocytes, and present histocompatibility, which means express MHC class I (HLA-ABC) and not express MHC class II (HLA-DR). The results showed that the cells isolated from Wharton’s jelly meet all the requirements pre-established by the International Society for Cell Therapy to be considered MSCs [22,23,24].

In addition to the differentiation into mesodermal lineages, the ability of MSCs to differentiate into other cell types, such as epithelial, neuronal, muscle, liver, heart, and lung cells, has already been proven [19,20]. In this study, it was possible to adapt a pulmonary differentiation protocol to differentiate WJ-MSC into lung epithelial cells, resulting in a lung organoid with secretory cells, ciliated cells, AT1, and AT2, which, until now, has only been achieved from induced pluripotent stem cell (iPSCs) and embryonic stem cell (ESCs) WJ-MSC [15,17,25,26,27]. The adaptation was performed by adding a WJ-MSC spheroid formation before lung differentiation. The spheroids simulate the embryonic bodies formed by iPSCs and ESCs, allowing differentiation into a lung organoid. This spheroid formation was developed by WJ-MSC cultivating on the natural functional biopolymer matrix (NFBX) membrane, a natural polymer membrane that induces mechanotransduction regulated by YAP/TAZ proteins, which are also associated with lung development [28,29,30].

After the spheroid formation, inducers that mimic cell signaling during lung organogenesis were used, resulting in the differentiation into organoids composed of secretory cells, ciliated cells, AT1, and AT2 [15,17,20,25,31]. Differentiation was confirmed by the presence of TTF1/NKX2.1, CD74, KRT18, SFTPC, AQP5, GRAMD2A, SCGB1A1, and MUC5AC markers both through RT-PCR and immunocytochemistry. Although the genes were expressed in both undifferentiated and differentiated cells in RT-PCR analysis, the quantification of the bands demonstrated that they are more expressed in the differentiated cells compared with the undifferentiated cells. Additionally, the proteins were not present in the undifferentiated WJ-MSC in immunocytochemistry analysis, indicating no translation, while in the differentiated lung cells, it was possible to observe the presence of the proteins, which confirms the differentiation.

The NKX2.1 gene, also known as TTF1, is the first known gene locus that is activated in primordial lung endoderm cells. It encodes a transcription factor that forms a regulatory loop with the GRHL2 gene, coordinating the morphogenesis and differentiation of lung epithelial cells, and its presence indicates the beginning of pulmonary differentiation [15,17,25,26].

AT2 characterization and the expression of SFTPC and CD74 were evaluated. SFTPC is the gene that codes for surfactant protein C, the main surfactant protein expressed only by AT2 and the main marker of this cell type [15,18,25,32,33]. CD74, also known as invariant chain protein, is a class II MHC-associated protein found in several cell types, although, in the lung, CD74 is associated only with AT2. In addition, the expression of CD74 indicates that the cells have differentiated, considering that undifferentiated MSCs do not express MHC class II. It was proven that the isolated cells were negative for HLA-DR [33,34,35,36].

For the presence of AT1, the markers AQP5 and GRAMD2AD were analyzed. The best-characterized marker of AT1 is aquaporin 5, a water channel protein expressed on the apical surface of the cells, encoded by the AQP5 gene. However, it is not only expressed in the lung but also in salivary and lacrimal glands [17,18,27,37]. On the other hand, GRAMD2A is a gene expressed only in AT1 and uterine tissue, and, therefore, the presence of both AQP5 and GRAMD2A simultaneously is used to identify AT1 [17].

To identify ciliated cells, cytokeratin 18, an intermediate filament cytoskeletal protein expressed in columnar cells by the KRT18 gene, was used as a marker [33,38,39,40,41]. Furthermore, SCGB1A1 and MUC5AC were evaluated for secretory cells because SCGB1A1 is the gene responsible for secretoglobulin. At the same time, MUC5AC is the gene responsible for producing mucin 5AC, which represents 95% of the total secretion of mucin in the airway epithelium [42,43,44].

After lung differentiation, the organoids were dissociated with collagenase, and the cells were mixed with sodium alginate and gelatin bioink for 3D bioprinting. Alginate is one of the main bioinks used for the 3D bioprinting of lung cells, and in most cases, it is mixed with gelatin [11,16,37,45]. Bioprinting was performed on an extrusion bioprinter, the most-used type of bioprinter, and is composed of a syringe, a nozzle, a pressure system, and a stage, which can move in three orthogonal planes. During bioprinting, the bioink is deposited on the stage by a dispensing head under the control of a computer [46,47,48]. The advantages of this method are low cost, flexibility, and speed, and the fact that it is compatible with a wide variety of bioinks, it can incorporate computer-aided design (CAD) software, and it is possible to print high cell densities. However, extrusion-based bioprinters have limited resolution and bioprinting is strongly linked to material crosslinking. It has lower cell viability than other printing techniques due to shear stress in the nozzle tip wall [49,50].

Despite the disadvantages of the extrusion-based bioprinter compared to other types, our 3D bioprinted lung model was well defined without deformations. Cell viability was also not impaired by the printing mode, remaining above 90% in all cases, both with undifferentiated WJ-MSC and differentiated lung cells. In addition, the cells remained with high viability, above 90%, 24, 48, and 72 h after printing, proving the biocompatibility of the bioink. Furthermore, the structure presented a homogeneous cell concentration, meaning that there was no precipitation of cells when the bioink was placed in the syringe. Precipitation would occur if the bioink were too liquid because its viscosity prevents that, demonstrating this bioink is suitable for 3D printing. The differentiated cells also continued to express all the pulmonary markers used in the immunocytochemistry after the 3D printing, confirming that the structure can be used as an in vitro 3D lung tissue model.

In previous studies, 3D lung models were bioprinted with A549 cells, an adenocarcinoma cell line, which is not the best alternative for studying drug toxicity in normal cells [11,14,45,49]. This study presents a healthy model with four cell types of the pulmonary epithelium. The lung epithelium is a monolayer, and the cells interact not only with each other, but also with endothelial cells for the air exchange. Despite the advantageous features of the model, there are still some limitations in terms of mimicking the natural situation. Further research and improvements are needed for a co-culture 3D structure with lung and endothelial cells.

## 4. Materials and Methods

### 4.1. Isolation and Culture of WJ-MSCs

The Research Ethics Committee of Pequeno Príncipe Faculty approved this study, which was numbered 3.288.245 (26 April 2019). The collections were completed at term, shortly after the placenta was discharged. Four human umbilical cord samples were collected and used in this study in triplicate. WJ-MSCs isolation was performed by the explant method. Briefly, the human umbilical cord samples were stored in a 50 mL tube with phosphate-buffered saline (PBS) (Sigma-Aldrich^®^, St. Louis, MO, USA) containing 3% penicillin/streptomycin (P/S) (300 UI/mL penicillin and 0.3 mg/mL streptomycin) (Sigma-Aldrich^®^, St. Louis, MO, USA). Umbilical cord processing was performed 4 h after collection in a laminar flow hood. Each sample was washed extensively with phosphate-buffered saline (PBS) (Sigma-Aldrich^®^, St. Louis, MO, USA) containing 3% penicillin/streptomycin (P/S) (300 UI/mL penicillin and 0.3 mg/mL streptomycin) (Sigma-Aldrich^®^, St. Louis, MO, USA). The blood vessels were removed, and the umbilical cord was minced into small fragments. These fragments were attached to a 75 cm^2^ culture flask and incubated for 10 min at 37 °C and 5% CO_2_ (Appendix A). Afterwards, a standard culture medium composed of Dulbecco’s Modified Eagle’s Medium/Nutrient Mixture F-12 (DMEM/F12) (Sigma-Aldrich^®^, St. Louis, MO, USA), 10% fetal bovine serum (FBS) (GIBCOTM Life Technologies/Thermo^®^, Waltham, MA, USA), and 1% P/S was added to the flask. The cells were cultured until they reached 80% confluence [50,51].

### 4.2. Characterization of WJ-MSCs

#### 4.2.1. Flow Cytometry

After trypsinization, a minimum of 1 × 10^6^ cells were resuspended in 1 mL PBS with 5% human albumin (HA) (Sigma-Aldrich^®^, St. Louis, MO, USA). The cell suspension was distributed in four cytometry tubes, and the conjugated antibodies were added according to Table 1. The tubes were vortexed and incubated in the dark for 15 min. After incubation, 400 μL of PBS with 5% HA was added, and the tubes were vortexed then centrifuged for 5 min at 600G. After centrifugation, the supernatant was discarded, and the cells were resuspended in 100 μL of PBS with 5% HA, and 5 μL of 7-AAD (7-amino actinomycin D) (Becton Dickinson^®^, Franklin Lakes, NJ, USA), used for cell viability, was added to the specific tubes that were incubated for 5 min. After incubation, 400 μL of PBS with 5% HA was added to each tube on the flow cytometer (FACS Canto II; Becton Dickinson^®^, Franklin Lakes, NJ, USA), 10,000 cells were analyzed, and data analysis was performed using the InfinicytTM software: Flow Cytometry Software 1.6.0 (Cytognos S.L., Santa Marta de Tormes, Spain). The gating strategy excluded non-viable cells (positive for the 7-AAD marker) and compared each marker with the isotypic control.

#### 4.2.2. Trilineage Assay

For the adipogenic differentiation, when WJ-MSc reached 70–80% confluence, the culture medium was supplemented with 0.5 μM dexamethasone (Sigma-Aldrich^®^, St. Louis, MO, USA), 0.5 mM isobutyl-methylxanthine (Sigma-Aldrich^®^, St. Louis, MO, USA), and 50 μM indomethacin (Sigma-Aldrich^®^, St. Louis, MO, USA). The cultivation with a differentiation medium was maintained for 14 days, and the medium was changed twice a week [51,52]. The accumulation of lipid vesicles was detected through oil red O staining (Sigma-Aldrich^®^, St. Louis, MO, USA).

The osteogenic and chondrogenic differentiations were induced using the StemPro^®^ Osteogenesis Differentiation Kit and StemPro^®^ Chondrogenesis Differentiation Kit, respectively, according to the manufacturer’s specifications (Thermo^®^, Waltham, MA, USA). The mineralization in the osteoblasts was evaluated by staining the cells with alizarin red (Sigma-Aldrich^®^, St. Louis, MO, USA), and the production of proteoglycans in the chondroblasts was detected through staining with alcian blue.

### 4.3. Pulmonary Differentiation

The first step for lung differentiation was spheroid formation. To do this, the cells were cultured on plates coated with NFBX membrane. The membrane comprises polyisoprene (C5H8), the main component of natural rubber, diluted with water 1:2 (*v*/*v*). The solution was used to cover the wells (0.5 mL/cm^2^), and polymerization occurred by keeping the plates at room temperature for at least 12 h. After polymerization, the plates were exposed to UV light for 2 h for sterilization. Then, the membranes were hydrated with culture media for 24 h. Subsequently, 20 μL of a WJ-MSC solution (2 × 10^4^ cells/mL) was seeded on the membrane, incubated for 20 min, and, after incubation, the culture medium was added to each well. The medium was changed twice a week until spheroid formation [53].

The spheroids were collected with a 1000 μL micropipette and crushed with a 100 μL micropipette to form small aggregates suspended in the medium. Aggregates were inoculated into a 48-well plate. This plate was shaken to ensure the homogenization of the aggregates in the well, and incubated in an oven at 37 °C and 5% CO_2_ for two days. After these two steps, the cells were cultured with the appropriate medium following Table 2, resulting in lung organoids after 70 days [15].

### 4.4. Characterization of Pulmonary Cells

#### 4.4.1. Immunocytochemistry

After the pulmonary differentiation, the lung organoids were characterized for the expression of CD74 (invariant chain protein), SFTPC (surfactant protein C), and AQP5 (aquaporin 5), which are expressed in AT1 and AT2, KRT18 (cytokeratin 18), which is an intermediate filament in ciliated cells, and SCGB1A1 (secretoglobin family 1A member 1) and MUC5AC (mucin 5AC), which are secretions produced by secretory cells. The lung organoids were fixed with 4% paraformaldehyde and permeabilized with 0.1% Triton X-100 (Sigma-Aldrich^®^, St. Louis, MO, USA). Afterwards, the primary antibodies were added, and the cells were incubated overnight at 4 °C. Then, the solutions with the primary antibodies were discarded, and the cells were washed with PBS. They were incubated for 1 h at room temperature with the secondary antibody without light. For the next step, the cells were washed, and 1 μg/mL of Hoechst 33258 (Invitrogen^®^, Carlsbad, CA, USA) diluted in PBS was used to identify the nucleus [54]. High-throughput fluorescence microscopy was used to acquire the images (In Cell Analyzer 2000, GE Healthcare^®^, Chicago, IL, USA). The dilution of each antibody is described in Table 3.

#### 4.4.2. Qualitative Reverse Transcription–Polymerase Chain Reaction (RT-PCR)

For the RT-PCR, the RNA of the lung organoids was extracted with the kit “PureLink TM RNA Mini Kit” (Invitrogen^®^, Carlsbad, CA, USA), following the instructions of the manufacturer. The RNA samples were treated with DNAse I to remove residual genomic DNA. The complementary strand of DNA (cDNA) was produced following the instructions of the “High-Capacity cDNA Reverse Transcription Kit” kit (Invitrogen^®^, Carlsbad, CA, USA). The PCR reaction was performed using a mix composed of 10× buffer, 10 mM dNTP, 10 µM forward primer, 10 µM reverse primer, 5 U/µL TAQ, and ultrapure water (all reagents were obtained from Sigma Aldrich^®^, St. Louis, MO, USA). The cDNA sample was added to the mix at a ratio of 2 μL of cDNA to 23 μL of the mix. The primer sequences (forward and reverse), the molecular weight of the amplified material, and the annealing temperature are described in Table 4. Sodium agarose gel electrophoresis was used to visualize the results obtained through RT-PCR, and ImageJ was used to quantify the bands. Statistical analyses were performed using the GraphPad Prism 5 software. Data normality was tested using the Shapiro–Wilk and two-way ANOVA statistical tests.

### 4.5. 3D Bioprinting

The bioink was prepared with sodium alginate (Sigma-Aldrich^®^, St. Louis, MO, USA) and gelatin (Sigma-Aldrich^®^, St. Louis, MO, USA). The gelatin (4% *w*/*v*) was dissolved in a saline solution with 1% P/S at 40 °C for 1h using a magnetic stirrer. Afterwards, the sodium alginate (3.25% *w*/*v*) was added to the gelatin solution and mixed for 2 h using a magnetic stirrer at 40 °C. The bioink was stored at 4 °C until use, and prior to adding the cells, the bioink was sterilized for 2 h in UV light [16].

For the bioprinting, the lung organoids were treated with Collagenase II (Sigma-Aldrich^®^, St. Louis, MO, USA) for cell dissociation. Then, they were mixed with the bioink at a concentration of 1 × 10^4^ cells/mL. The bioink with cells was stored in a 5 mL syringe attached to the bioprinter. A bioprinter from 3D Biotechnology Solutions (3DBS^®^—Campinas, SP, Brazil) was used to print the structures. After printing, cross-linking was performed with a 10 mM calcium chloride solution (Sigma-Aldrich^®^, St. Louis, MO, USA). The calcium chloride solution was removed, the culture medium was added, and the structures were incubated at 37 °C and 5% CO_2_.

### 4.6. Characterization of the 3D Lung Scaffold

#### 4.6.1. Live/Dead Assay

The 3D constructs were incubated for 24, 48, and 72 h for cell viability evaluation. After incubation, markers from the live/dead viability kit (Invitrogen^®^, Carlsbad, CA, USA) were added to the culture medium: for live cells, calcein at 0.3 μM; for dead cells, ethidium homodimer at 0.6 μM; and Hoechst 33342 (Invitrogen^®^, Carlsbad, CA, USA) 2 μg/mL for nuclei. The plates were incubated at 37 °C, 5% CO_2_ for 30 min, and images were acquired using high-throughput microscopy (In Cell Analyzer 2000, GE Healthcare^®^, Chicago, IL, USA). Statistical analyses were performed using the GraphPad Prism 5 software. Data normality was tested using the Shapiro–Wilk test and the two-way ANOVA statistical test.

#### 4.6.2. Immunocytochemistry

After the 3D printing, the constructs were incubated for 48 h with a culture medium. After incubation, they were fixed with 4% paraformaldehyde (Sigma-Aldrich^®^, St. Louis, MO, USA) and 10 mM of calcium chloride (Sigma-Aldrich^®^, St. Louis, MO, USA). The permeabilization and immunostaining were performed as described above for the lung organoids.

## 5. Conclusions

This study successfully demonstrated the fabrication of a bioprinted 3D lung alveolus model derived from human WJ-MSC differentiated into lung cells. This model was designed to perform cytotoxicity tests of drugs against COVID-19. However, it may be used for other pulmonary diseases, reducing the use of animals for drug testing and obtaining a model closer to the reality of the human organism.

The isolation, cultivation, and characterization of WJ-MSC were presented, in addition to their differentiation into lung cells, which was confirmed using characterization by immunocytochemistry and RT-PCR. This study is the first report of MSC differentiation into pulmonary organoids with four cell types of the pulmonary epithelium. The spheroid formation was only possible through cultivation on the NFBX membrane.

The differentiated cells were used for 3D bioprinting in an in vitro lung model using sodium alginate and gelatin bioink. The 3D structure remained, with viable cells expressing proteins of AT1, AT2, ciliated, and secretory cells. This is also the first report of 3D bioprinting with four cell types of the pulmonary epithelium.

## Figures and Tables

**Figure 1 ijms-24-05852-f001:**
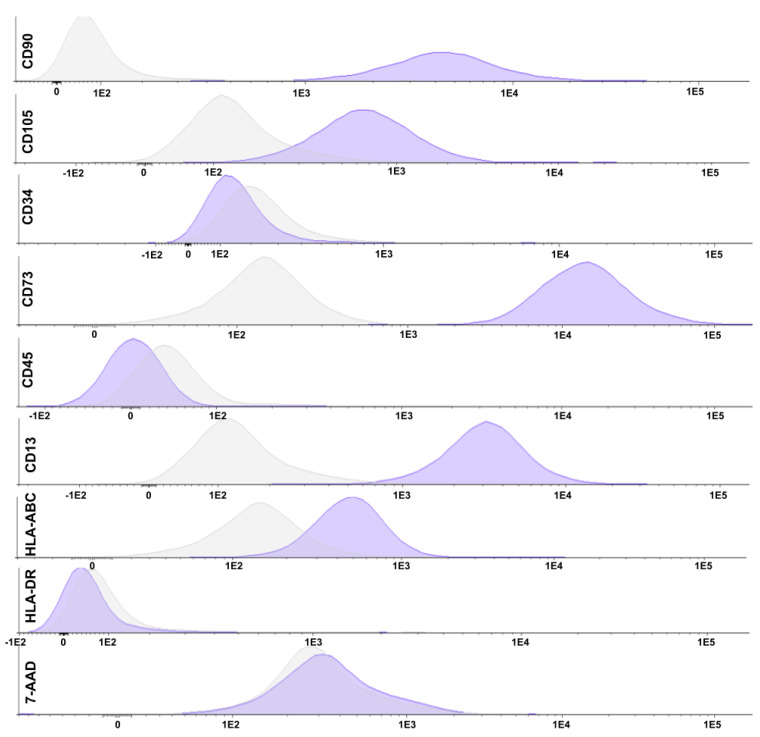
Flow cytometry histograms. The histograms represent the cytometry of the WJ-MSC (sample C6) for the markers CD34, CD45, CD73, CD90, CD105, HLA-ABC, and HLA-DR, respectively (purple), as well as the isotypic control (gray). 1E2= 1 × 10^2^, 1E3= 1 × 10^3^, 1E4= 1 × 10^4^ and, 1E5= 1 × 10^5^.

**Figure 2 ijms-24-05852-f002:**
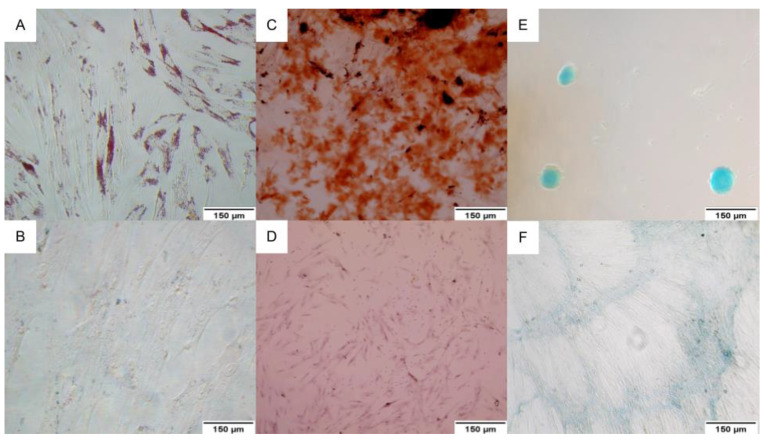
Trilineage differentiation. (**A**) WJ-MSC differentiated into adipocytes presenting lipid vacuoles stained with oil red O; (**B**) Control of undifferentiated WJ-MSC stained with oil red O; (**C**) WJ-MSC differentiated into osteoblasts showing mineralization stained with alizarin red; (**D**) Undifferentiated WJ-MSC stained with alizarin red; (**E**) WJ-MSC differentiated into chondroblasts with a production of proteoglycans stained with alcian blue; (**F**) Undifferentiated WJ-MSC stained with alcian blue. Image obtained through inverted optical microscopy, 100× (Axio Vert. A1, Zeiss, Oberkochen, BW, Germany).

**Figure 3 ijms-24-05852-f003:**
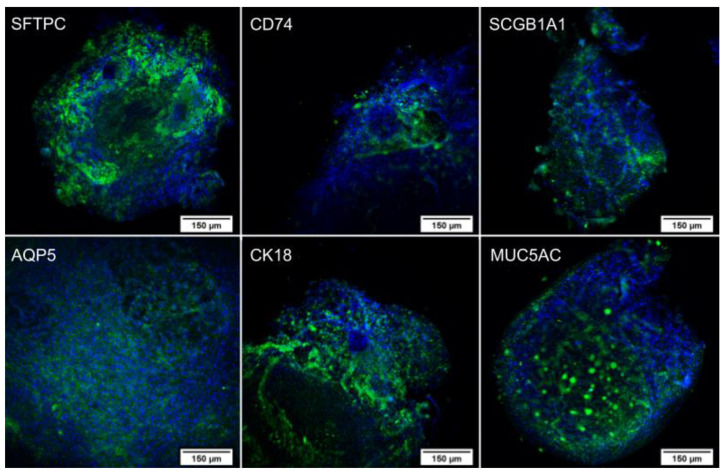
Immunocytochemistry of lung organoids (sample C10) with DAPI marking the nuclei in blue and FITC labeling the specific proteins in green. Images obtained using a high-throughput microscope In Cell Analyzer 2000 (GE Healthcare^®^, Chicago, IL, USA).

**Figure 4 ijms-24-05852-f004:**
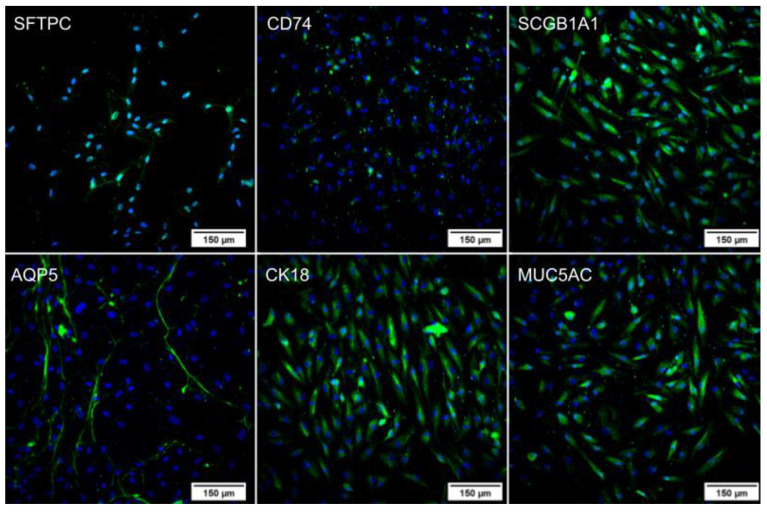
Immunocytochemistry of lung cells (sample C6) with DAPI labeling the nuclei in blue and FITC marking specific proteins in green. Images obtained using a high-throughput microscope In Cell Analyzer 2000 (GE Healthcare^®^, Chicago, IL, USA).

**Figure 5 ijms-24-05852-f005:**
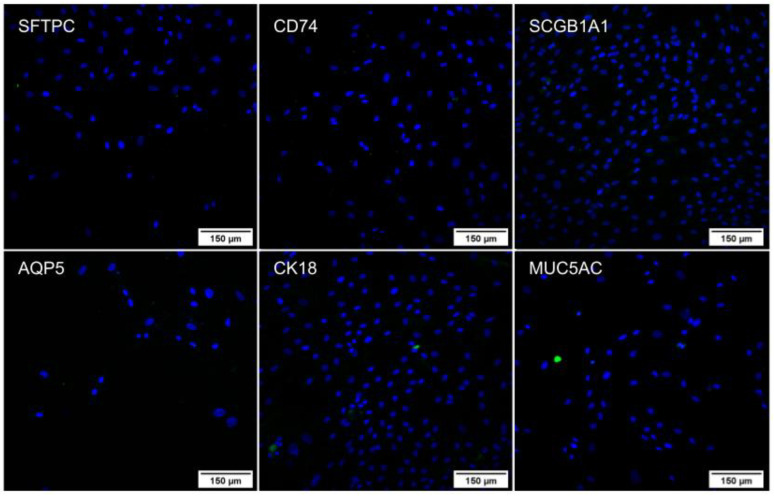
Immunocytochemistry of undifferentiated WJ-MSC (sample C6) with DAPI marking the nuclei in blue and FITC marking the specific proteins in green. Images obtained using a high-throughput microscope In Cell Analyzer 2000 (GE Healthcare^®^, Chicago, IL, USA).

**Figure 6 ijms-24-05852-f006:**
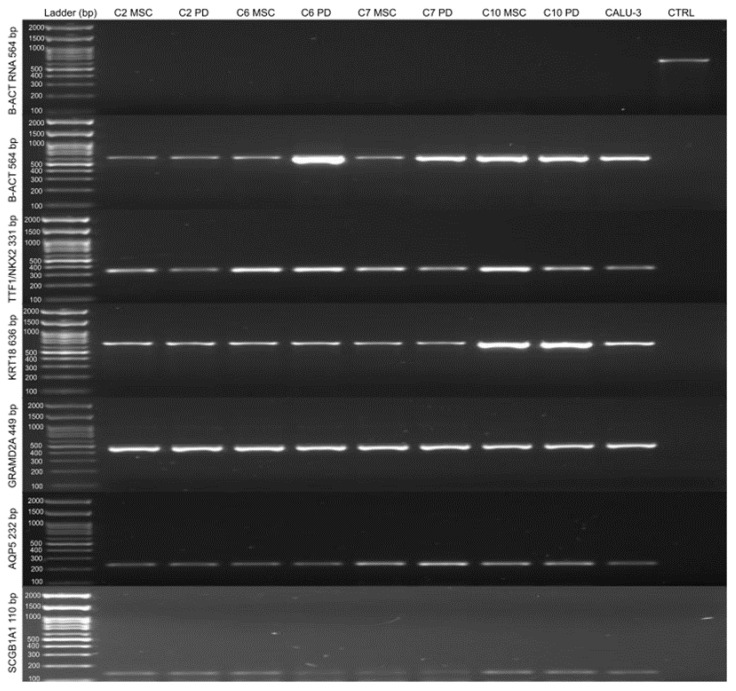
RT-PCR for lung cell genes. RT-PCR for B-ACT (564 bp) in a DNAse-treated RNA sample, B-ACT (564 bp) in a cDNA sample, TTF1/NKX2.1 (331 bp), KRT18 (636 bp), GRAMD2A (449 bp), AQP5 (232 bp), and SCGB1A1 (110 bp) genes, respectively, from top to bottom. RT-PCR was performed in four samples (C2, C6, C7, and C10), both undifferentiated (CT) and after lung differentiation (DP); as a positive control, the Calu-3 cell lineage was used, and as a negative control, a reaction was performed without addition of DNA. The molecular weights range from 100 bp to 2000 bp; from 100 to 1000, each hundred is indicated.

**Figure 7 ijms-24-05852-f007:**
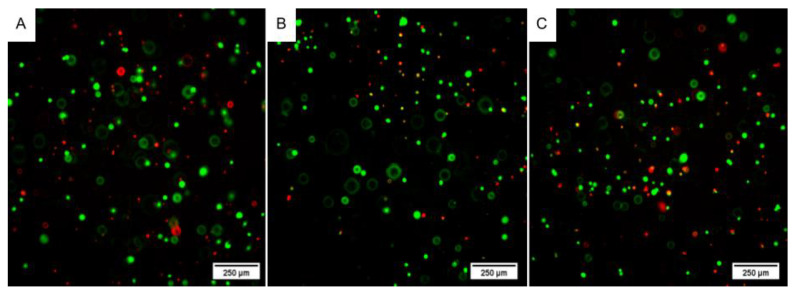
3D bioprinting with WJ-MSC. (**A**) Cell viability in the 3D structure after 24 h; (**B**) Cell viability in the 3D structure after 48 h; (**C**) Cell viability in the 3D structure after 72 h. The cells in green were viable, and in red, non-viable as dead. Images obtained using a high-throughput microscope In Cell Analyzer 2000 (GE Healthcare^®^, Chicago, IL, USA).

**Figure 8 ijms-24-05852-f008:**
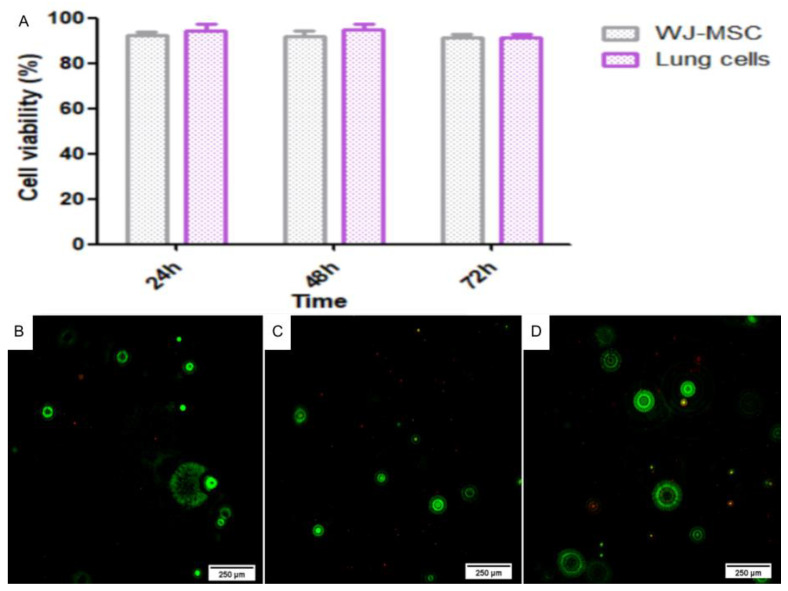
Live/dead assay of the lung 3D bioprinting. (**A**) Histograms indicating the viability of WJ-MSC (gray) and lung cells (purple) in the 3D structure after 24, 48, and 72 h of bioprinting. (**B**) Lung cell viability in the 3D structure after 24 h, with green live cells and red dead cells with 20× magnification. (**C**) Lung cell viability in the 3D structure after 48 h, with green live cells and red dead cells with 20× magnification. (**D**) Lung cell viability in the 3D structure after 72 h, with green live cells and red dead cells with 20× magnification. Images obtained using a high-throughput microscope In Cell Analyzer 2000 (GE Healthcare^®^, Chicago, IL, USA).

**Figure 9 ijms-24-05852-f009:**
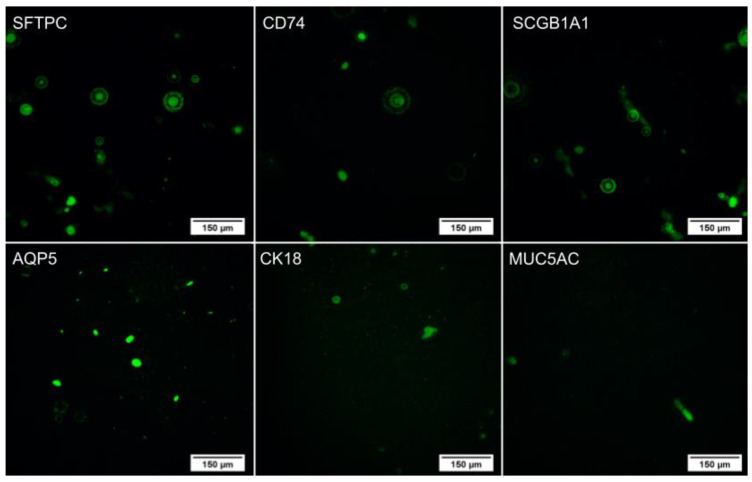
Immunocytochemistry of the 3D lung structures with FITC labeling specific proteins in green. Images obtained using a high-throughput microscope In Cell Analyzer 2000 (GE Healthcare^®^, Chicago, IL, USA).

**Table 1 ijms-24-05852-t001:** Cytometry panel.

Tube	Content
1	Cells without markers
2	Isotypic control
3	CD90 FITC/CD105 PE/7-AAD PERCP/CD34 PE-CY7/CD73 APC/CD45 APC-CY7
4	HLA-DR FITC/CD13 PE/7-AAD PERCP/CD34 PE-CY7/HLA-ABC APC/CD45 APC-CY7

The antibodies were obtained from Becton Dickinson^®^ (Franklin Lakes, NJ, EUA).

**Table 2 ijms-24-05852-t002:** Lung differentiation culture medium.

Day	Medium
1	RPMI + 100 ng/mL activin A
2	RPMI + 0.2% (vol/vol) FBS + 100 ng/mL activin A
3	RPMI + 2% (vol/vol) FBS + 100 ng/mL activin A
4	RPMI + 2% (vol/vol) FBS + 100 ng/mL activin A
5–10 (changing medium every day)	DMEM/F12 + 1× N2 + 1× B27 + 1× L-glutamine + 1× P/S + 10 mM HEPES buffer + 10 µM SB431542 + 200 ng/mL Noggin + 1 µM SAG + 500 ng/mL FGF4 + 2 µM CHIR99021
10–70 (changing medium twice a week)	DMEM/F12 + 1× N2 + 1× B27 + 1× L-glutamine + 1× P/S + 10 mM HEPES buffer + 1% (*v*/*v*) FBS e 500 ng/mL FGF10

RPMI (Sigma-Aldrich^®^, St. Louis, MO, USA), FBS (GIBCOTM Life Technologies/Thermo^®^, Waltham, MA, USA), all the supplements were from Peprotech^®^ (Ribeirão Preto, SP, Brazil).

**Table 3 ijms-24-05852-t003:** Antibodies for immunocytochemistry.

Antibody	Dilution	Code
AQP5	2 μg/mL	HPA065008
CD74	1:100	SAB521932
KRT18	1:100	C8541
MUC5AC	1:100	M5293
SCGB1A1	1: 100	SAB1411381
SFTPC	1:100	ZRB1496
Anti-mouse secondary antibody with FITC	10 μg/mL	F7512
Anti-rabbit secondary antibody with FITC	10 μg/mL	F0382

The antibodies were from Sigma Aldrich^®^ (St. Louis, MO, USA). FITC, Fluorescein isothiocyanate.

**Table 4 ijms-24-05852-t004:** Primers for RT-PCR.

Gene	Forward	Reverse	MW (pb)	AT (°C)
B-ACT	CTGGGACGACATGGAGAAAA	AAGGAAGGCTGGAAGAGTGC	564	60
TTF1/NKX2.1	CGGCGCTTTCGGAGGGAATA	TGTAACACCTGCTTCCTCGTC	331	62
KRT18	AAAGCCTGAGTCCTGTCCTT	CCAGCTGCAGTCGTGTGATA	636	60
GRAMD2A	ATGACCGCTTTAAGCCGGAG	ATGGCCAGTCCATTGGGAAG	449	62
AQP5	TCCATTGGCCTGTCTGTCAC	CTTTGATGATGGCCACACGC	232	58
SCGB1A1	TCCACCATGAAACTCGCTGT	AGGAGGGTTTCGATGACACG	110	62

The primers were from Sigma Aldrich^®^ (St. Louis, MO, USA).

## Data Availability

Not applicable.

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
