# Peer review of "Three-Dimensional Bioprinting of an In Vitro Lung Model"

_ijms, 2023, doi:10.3390/ijms24065852_

Round 1

Reviewer 1 Report

The introduction is incomplete and does not cover the previous works in this domain.

The innovation of this work compared to previous studies is missing.

Materials and methods are missing. Materials/ machines and all the information related to machines and materials missing in this section.

What are CD13, CD73, etc. there is no information on what these terms refer to.

Nomenclature must be added to the paper.

There is no explanation of sample preparation, conditions, or even a photo of samples.

The discussion of the results is incomplete.

The order and organization of the paper are wrong. The order is as follows: Introduction, results, discussion, materials and methods and again results. It is the first time I see an article with materials and methods after results and discussion. Please reorder the paper and add materials and methods after the introduction.

The conclusion needs to be improved.

Author Response

RESPONSE TO REVIEWER REPORT 1

  1. The introduction is incomplete and does not cover the previous works in this domain. Response:We are grateful for your carefully read; we have included more of the earlier works as “state of the art “in the introduction as added a paragraph with previous lung bioprinting work. We believe these corrections ameliorate the background.

Response: Thank you very much for your comments.

  1. The innovation of this work compared to previous studies is missing. Response:We apologize for missing these comparisons; they were emphasized in the discussion and included.

  1. Materials and methods are missing. Materials/ machines and all the information related to machines and materials missing in this section.Response: Thanks for your comments; the materials /Machines and their Registered symbol were included. Now in all materials /Machines, it was marked.

  1. What are CD13, CD73, etc. there is no information on what these terms refer to. Response: We apologize that we do not explain this information. The Cluster of Differentiation, abbreviature CD, is cell superficies’ markers identifying cell surface antigens. Each number represents a specific marker of the cells that represents a gene and protein phenotype. We have included it in the main text.

Nomenclature must be added to the paper. Response: We included the name of each cell superficies marker protein in the main text. CD90: Thy-1; CD105: Endoglin; CD73: Ecto-5-prime-nucleotidase; CD13: Aminopeptidase-N; CD45: Leukocyte common antigen; CD34: hematopoietic progenitor cell antigen. All other nomenclature and terminology were included in the main text.

  1. There is no explanation of sample preparation, conditions, or even a photo of samples. Response: Thank you for your comments. More details on sample collection and isolation umbilical cord proceedings included to isolate the cells from Wharton Jelly were added in topic 4.1 and added Supplementary Figure 2 with the sample images.

  1. The discussion of the results is incomplete.Response: We apologize. We rewrite it, and we hope that you appreciate it now.

  1. The order and organization of the paper are wrong. The order is as follows: Introduction, results, discussion, materials, and methods and again results. It is the first time I see an article with materials and methods after results and discussion. Please reorder the paper and add materials and methods after the introduction. Response: We are following the Template of this submitted journal in this order Introduction, Results, Discussion, Materials and Methods, and Conclusions. Nevertheless, the conclusions needed clarification with the Results. We improved these findings. 

  1. The conclusion needs to be improved. Response: We ameliorated the conclusions. You can read in the main text.

We made English corrections and revised the manuscript carefully.

We sincerely thank you for your constructive criticisms and valuable comments, which helped revise the manuscript.

Corresponding author,

Reviewer 2 Report

The work described a strategy to obtain lung spheroids starting from differentiated mesenchymal stem cells (MSCs). The authors briefly characterized by cytofluorimetric analysis the derived MSCs and then, they differentiated them into three lineages (i.e. adipocytes, osteoblasts and chondroblast) to confirm the differentiation capacity. After lung spheroids were obtained and characterized by immunocytochemistry, they were included into an alginate-gelatin based matrices to form a printable bioink. Post-printing viability of loaded MSCs and lung cells was, then, evaluated. Although the novelty in the use of lung spheroid with extrusion-based bioprinter, the results are not well presented and the potential use of the model as drug testing platform was suggested but not investigated. Further, some claimed conclusion are not fully supported by results.

Major concerns

In section 2.2.1, immunocytochemistry analysis was performed to assess cell differentiation. However, no negative controls (i.e. not-differentiated cells) were shown and since, their importance to clearly demonstrate the induction of differentiation of MSCs towards lung epithelium phenotype, they must be added. Then, it is not specified the time point in which that stainings were performed.

In section 2.2.2, the authors stated that the expression of gene related to secretory cells (i.e. AT1 and AT2) is equal in undifferentiated and differentiated cells. To state that, qRT-PCR analysis seems to be more suitable technique than RT-PCR. Why was RT-PCR preferred? Further, if no differences were detected how the authors demonstrated that the differentiation really happened? Again, the different time points in which the authors performed the analysis were not clearly described.

In Figure 5, the molecular weights indicated by the DNA ladder are not shown. It could be useful to indicate the range.

In section 2.3., the authors stated that “the structures were printed according to the model without deformations” but neither the model nor the whole printed sample were shown. Then, strategies to avoid cell precipitation into the syringe were not described or shown.

In section 2.3., the authors mentioned a Live/Dead assay performed at three different time points, but, only one image (Figure 6) was shown without any information regarding the time point. Further, the mentioned quantification (i.e. 92.55%, 92.38% and 91.66% in 24, 48 and 72 hours) probably is the one shown in Figure 7, not 6 as was written.   

In Figure 7A, the stained cells are not clearly visible. Then, all the time points are not shown to demonstrate the cell viability obtained.

In section 2.4.2 is described the characterization of printed structures. Particularly, the cells were stained with antibodies that recognize protein associated to epithelial differentiation. However, unlike in Figure 3 and 4, cells in Figure 8 were not counterstained with nuclear dye (DAPI), so, it is impossible to understand the localization of the cells and, in turns, of the stained proteins. Further, the quality seems quite different from the previously showed staining.

In the abstract and in the introduction, the model is proposed as a platform for drug testing, but, analysis to demonstrate this potential have not been performed.

Minor comments

Figure quality must be improved. For instance, in Figure 1, values in x-axis are not visible and in Figure 2E and F, cells are not clearly detectable. In general, values of scale bar in the images are not always clear.

Many typos should be corrected.

Author Response

RESPONSES TO REVIEWER REPORT 2

Major concerns

  1. In section 2.2.1, immunocytochemistry analysis was performed to assess cell differentiation. However, no negative controls (i.e. not-differentiated cells) were shown and since, their importance to clearly demonstrate the induction of differentiation of MSCs towards lung epithelium phenotype, they must be added. Then, it is not specified the time point in which that stainings were performed. Response:  I apologize for our omission. Immunocytochemistry of the undifferentiated cells was performed, and it was negative for proteins; the images were added to figure X and adjusted in the main text.

  1. In section 2.2.2, the authors stated that the expression of gene related to secretory cells (i.e. AT1 and AT2) is equal in undifferentiated and differentiated cells. To state that, qRT-PCR analysis seems to be more suitable technique than RT-PCR. Why was RT-PCR preferred? Further, if no differences were detected how the authors demonstrated that the differentiation really happened? Again, the different time points in which the authors performed the analysis were not clearly described. Response:Due to the negative immunocytochemistry of the undifferentiated cells, we concluded that there was gene transcription but not translation. On the other side, in differentiated cells, there is the presence of proteins, which indicates that there has been differentiation. Therefore, we chose not to proceed with qRT-PCR.

  1. In Figure 5, the molecular weights indicated by the DNA ladder are not shown. It could be useful to indicate the range. Response: Molecular weights range from 100 bp to 2000 bp, with 100 to 1000 indicated per hundred. The image in Figure X has been adjusted.

  1. In section 2.3., the authors stated that “the structures were printed according to the model without deformations” but neither the model nor the whole printed sample were shown. Then, strategies to avoid cell precipitation into the syringe were not described or shown. Response: The print model and bioprinted structure were added as Supplementary Figure 1, and it explained in the discussion that the viscosity of the bioink prevents precipitation.

  1. In section 2.3., the authors mentioned a Live/Dead assay performed at three different time points, but, only one image (Figure 6) was shown without any information regarding the time point. Further, the mentioned quantification (i.e. 92.55%, 92.38% and 91.66% in 24, 48 and 72 hours) probably is the one shown in Figure 7, not 6 as was written. Response: We apologize for our mistake in presenting these results. The figure was changed with the three beats of Live/Dead, and the numbering was corrected in the text. Thank you very much for your reading with attention to our manuscript.

  1. In Figure 7A, the stained cells are not clearly visible. Then, all the time points are not shown to demonstrate the cell viability obtained. Response: Figure 7A was changed. It was removing the view of the entire structure and placing all the time points with the highest increase magnification to demonstrate the cell viability.

  1. In section 2.4.2 is described the characterization of printed structures. Particularly, the cells were stained with antibodies that recognize protein associated to epithelial differentiation. However, unlike in Figure 3 and 4, cells in Figure 8 were not counterstained with nuclear dye (DAPI), so, it is impossible to understand the localization of the cells and, in turns, of the stained proteins. Further, the quality seems quite different from the previously showed staining. Response: To understand, we would like to explain. In Figure 8, the immunocytochemistry carried out in the 3D structure. For this purpose, the cells were dissociated from the spheroids and were not adhered to the plastic to explain the morphology change. Furthermore, as it is a 3D structure, the overlapping of cells on the vertical axis hampered the quality of the images. It made it impossible to counterstain with nuclear dye (DAPI) because we did not have a confocal microscope.

  1. In the abstract and the introduction, the model is proposed as a platform for drug testing, but, analysis to demonstrate this potential have not been performed. Response:  The model is undoubtedly proposed as a platform for drug testing due to its three-dimensional architecture with four different types of alveolar cells, all types differentiated from human pluripotentcells, from WJMSCs in stratified arrangement composing an organoid structure. This 3D structure is ready to start drug testing. However, these tests will be done by specialists in cytotoxicity.

Minor comments

  1. Figure quality must be improved. For instance, in Figure 1, values in x-axis are not visible and in Figure 2E and F, cells are not clearly detectable. In general, values of scale bar in the images are not always clear. Response: All photos have been adjusted to improve their sharpness.

  1. Many typos should be corrected. Response: These types of errors and English grammatical revision were carefully done.

We made English corrections and revised the manuscript carefully.

We sincerely thank you for your constructive criticisms and valuable comments, which helped revise the manuscript.

Corresponding author,

Round 2

Reviewer 2 Report

Dear authors,

Although the main concerns about methodology and results were addressed, some errors are still present.

In Figure 1 the values on the x-axes are still not clearly visible and in Figure 2F the cells are not detectable.

In Figure 7 the new description that address my request to show the viability of all the time points analyzed is not associated to the right pictures. In the upload version the figure is still the one of the unrevised version.

Figure 8 has the same problem, new description was added but the pictures of all the time points are not present in the revised version.

Author Response

Dear Reviewer,

Second Revision,

Thank you very much for your attention to our manuscript. See attached the responses with figures about your considerations.

  1. In Figure 1 the values on the x-axes are still not clearly visible and in Figure 2F the cells are not detectable.

Response:

1a. It was increased the font size of the values on the x-axis of figure 1.

1b. Was changed figure 2F to another one in which the cells are more visible.

  1. In Figure 7 the new description that address my request to show the viability of all the time points analyzed is not associated to the right pictures. In the upload version the figure is still the one of the unrevised version.

Response: Unfortunately, we do not know what happened with both figures’ files. We have made the corrections and sent them before.  

  1. Figure 8 has the same problem; new description was added but the pictures of all the time points are not present in the revised version.

Response: We apologize; the same thing arrived in this figure 8.

We sincerely thank the reviewers for their constructive criticisms and valuable comments, which helped revise the manuscript.
